# Are You Stealing My Model? Sample Correlation for Fingerprinting Deep Neural Networks

**Jiyang Guan**[1,2], **Jian Liang**[1,2], **Ran He**[1,2*]

[1]NLPR &CRIPAC, Institute of Automation, Chinese Academy of Sciences, China
[2]School of Artificial Intelligence, University of Chinese Academy of Sciences, China
`guanjiyang2020@ia.ac.cn, liangjian92@gmail.com, rhe@nlpr.ia.ac.cn`

## Abstract

An off-the-shelf model as a commercial service could be stolen by model stealing attacks, posing great threats to the rights of the model owner. Model fingerprinting aims to verify whether a suspect model is stolen from the victim model, which gains more and more attention nowadays. Previous methods always leverage the transferable adversarial examples as the model fingerprint, which is sensitive to adversarial defense or transfer learning scenarios. To address this issue, we consider the pairwise relationship between samples instead and propose a novel yet simple model stealing detection method based on SAmple Correlation (SAC). Specifically, we present SAC-w that selects wrongly classified normal samples as model inputs and calculates the mean correlation among their model outputs. To reduce the training time, we further develop SAC-m that selects CutMix Augmented samples as model inputs, without the need for training the surrogate models or generating adversarial examples. Extensive results validate that SAC successfully defends against various model stealing attacks, even including adversarial training or transfer learning, and detects the stolen models with the best performance in terms of AUC across different datasets and model architectures. The codes are available at `https://github.com/guanjiyang/SAC`.

## 1 Introduction

Over the past years, Deep Neural Networks (DNNs) have played an important role in many critical fields, e.g., face recognition [1], medical diagnosis [2, 3] and autonomous driving [4]. As a popular option, the model owners always provide their models as a cloud service or client-sided software to the clients. But, training a deep neural network is costly and involves expensive data collection and large computation resource consumption, and thus, models trained for inference constitute valuable intellectual property and should be protected [5, 6]. However, model stealing attacks can steal the valuable model with only API access to the model owner's well-performed model (source model) [7], causing serious threats to the model owner's intellectual property (IP).

Model stealing attacks aim to illegally steal functionally equivalent copies of the source model with white-box or even black-box access to the model. As for the white-box case, the attacker can access all the inner parameters of the source model and evade the model owner's detection by source model modification such as pruning [8, 9] or fine-tuning the source model. By contrast, model extraction attack [10, 11] as a more powerful attack, has been proposed with the black-box access to the model. That is to say, the model extraction attack only requires model outputs rather than the inner parameters to steal the function of the source model, and thus is more threatening.

---

*Corresponding Author

36th Conference on Neural Information Processing Systems (NeurIPS 2022).

As model stealing has raised considerable concerns about model ownership, an increasing number of model IP protection methods have been proposed in the last few years. Generally, there are two categories to validate and protect the source model's IP, i.e., the watermarking methods [5, 12–19] and the fingerprinting methods [7, 20–23]. The watermarking methods use weight regularization [12–14] or backdoor inserting [5, 16, 17] during model training and leave the specific watermark in the model. However, they need to involve the model's training procedure and sacrifice the model's performance on the main task. A typical example is EWE [5] that witnesses a $4\%$ classification accuracy drop on CIFAR10. On the contrary, the fingerprinting methods make use of adversarial examples' transferability and identify the stolen models by adversarial examples' attack success rates on the suspect model. Since they do not involve in the training procedure, model fingerprinting does not influence the model's accuracy. Nevertheless, these adversarial example based fingerprinting methods are still sensitive to adversarial training [24]. They also take up a large amount of time for the model owner to train the surrogate models (models that the model owner trains using model extraction by themselves) and generate adversarial examples. In addition, with the changes in label space on transfer-learning, the adversarial examples' target label disappears, meaning that they also can not identify model stealing attacks with transfer learning techniques [25].

As stated above, existing fingerprinting methods, leveraging the suspect model's output as a point-wise indicator to detect the stolen models, are sensitive to adversarial training or transfer learning. To address this problem, we focus on the pair-wise relationship between the outputs and develop a new method called SAC. Intuitively, samples with similar outputs in the source model are more likely to also have similar outputs in the stolen models. In particular, we employ the correlation difference between the source model and the suspect model as the indicator to detect the stolen model. However, calculating correlation using all the samples from the defender's dataset will be influenced by the common knowledge shared by most models trained for the same task, on which most models will output the same label. To avoid it, we leverage the normal samples which are wrongly predicted by both the source and the surrogate models as the model input and propose to fingerprint using sample correlation with wrongly predicted samples (SAC-w). Furthermore, to reduce the needs for a large number of normal samples and save time consumption for the defender, we use CutMix Augmented samples directly to calculate the correlation difference (SAC-m), which does not need to train the surrogate models or generate adversarial examples. To verify the effectiveness of SAC-w and SAC-m, we investigate 5 types of attacks (i.e., fine-tuning, pruning, transfer learning, model extraction, and adversarial training), and compare the performance against these attacks across different model architectures and datasets.

Our main contributions are summarized as follows:

- We introduce sample correlation into model IP protection, and propose to leverage the correlation difference as the robust indicator to identify the model stealing attacks and provide a new insight into model IP protection.

- We introduce the wrongly-predicted normal samples and CutMix Augmented samples to replace adversarial examples as the model inputs, providing two robust correlation-based model fingerprinting methods.

- Extensive results verify that SAC is able to identify different model stealing attacks across different architectures and datasets with $AUC = 1$ in most cases, performing better than previous methods. Besides, SAC-m only takes up 4.45 seconds on the CIFAR10 dataset, greatly lowering the model owner's computation burden.

## 2 Related Work

Model stealing attacks greatly threaten the rights of the model owner. In general, we can summarize them in several categories as follows: (1) Fine-tuning [26]: The attacker updates the parameters of the source model using the labeled training data with several epochs. (2) Pruning [8, 9]: The attacker prunes less significant weights of the source model based on some indicators such as activation. (3) Transfer learning [25]: The attacker transfers the source model to some similar tasks and makes use of the source model's knowledge. (4) Model extraction [10, 11]: Because data labeling is costly and time-consuming, and there is a large amount of unlabeled data on the Internet, the attacker can steal the function of the source model using only the unlabeled same-distribution data. Different from the above attacks which need access to the inner parameters of the model, the model extraction attack

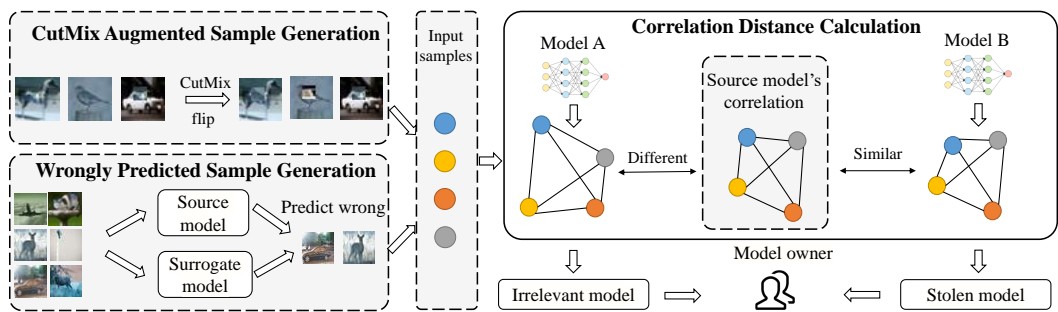

Figure 1: Correlation fingerprinting framework. We first generate CutMix Augmented samples or misclassified samples as model inputs, represented by colored balls. Then we calculate the correlation difference and any suspect model with a similar correlation will be recognized as a stolen model.

can steal the source model with only the source model's output. (5) Adversarial training [24]: The attacker can train models with both the normal examples and the adversarial examples, which help evade most fingerprinting detection.

Because of the threat of model stealing attacks, there are many model intellectual property (IP) protection methods having been proposed. Generally, there are two main categories to validate and protect the model IP, the watermarking methods and the fingerprinting methods. Watermarking methods usually leverage weight regularization [12–15] to put secret watermark in the model parameters or train models on triggered set to leave backdoor [16, 17] in them. However, these methods can not detect newer attacks such as model extraction which trains a stolen model from scratch [7, 11]. There are also some watermarking methods such as VEF [27] or EWE [5], which can survive during the model extraction. However, VEF needs a white-box access to the suspect model, limiting the scope of its application. Also, they all need to involve the training process, which sacrifices the model's accuracy [5, 18, 19], and in many critical domains, even 1% accuracy loss is intolerable [20].

Fingerprinting, on the contrary, utilizes the transferability of the adversarial examples and can verify models' ownership without participating in the models' training process, guaranteeing no accuracy loss. Lukas et al. [7] proposes conferrable adversarial examples to maximize adversarial examples' transferability to stolen models and minimize transferability to independently trained models (irrelevant models). Besides, ModelDiff [21], FUAP [22], DFA [23] leverage different kinds of adversarial examples such as DeepFool [28] and UAP [29], to fingerprint the source model. However, all these methods rely on adversarial examples and can be easily cleared out by adversarial defense such as adversarial training [24] or transfer learning. Furthermore, these methods usually need to train lots of surrogate models and irrelevant models with different model architectures to form well-established fingerprints, causing a great computation burden for the model owner. Besides, DeepJudge [30] proposes a unified framework to use different indicators to detect model stealing attacks in both the white-box and the black-box settings. Unlike previous methods, our method makes use of the correlation between samples rather than just the instance level difference in previous methods and can fingerprint models much faster and more robust with data-augmented samples instead of adversarial examples. Moreover, different from Teacher Model Fingerprinting [31], which uses the paired samples generated from the matching of the representation layers to detect the transfer learning attack, our method makes use of the correlation of independent samples and can detect more categories of model stealing attacks.

## 3 Proposed Method

### 3.1 Problem Definition

There are two parties in the model IP protection, the defender and the attacker. The defender is the model owner, who trains a well-performed model with a (proprietary) training dataset and algorithm [20]. The defender can deploy their well-trained models as a cloud service or client-sided software [20]. In cloud service setting, the attacker can only get the output of the model. On the contrary, in client-sided software setting, the attacker can get access to all the inner parameters of the models. The

attacker's goal is to use their same-distribution data to derive a stolen model with similar accuracy to the source model but evade the defender's detection. The defender's goal is to identify whether the suspect model is stolen from him. Because the attacker may deploy their stolen model on cloud, our IP protection is based on a black-box setting, where the defender can only get access to the output of the suspect model and can not have the knowledge of the stolen model's architecture.

## 3.2 Sample Correlation for Neural Network Fingerprinting

Previous fingerprinting methods only consider the point-wise consistency between the source model and the stolen model, and identify the stolen model by whether it classifies the same adversarial examples into the same wrong classes as the source model. However, they are not robust to adversarial training or transfer learning. For robustly detecting the stolen models, instead of the point-wise criteria, we focus on the pair-wise relationship between the outputs. Intuitively, two samples with similar outputs in the source model are more likely to have similar outputs in the stolen models. Thus, we introduce a correlation-based fingerprinting method SAC which makes use of the correlation consistency mentioned above to recognize the stolen models. Furthermore, to get rid of the influence of common knowledge shared by irrelevant models, we study how to find suitable samples in Section 3.3. An overview of our framework is shown in Figure 1.

Then, we introduce how to use sample correlation [32, 33] as the model fingerprint, let $O_{source} = \{o_1^{source}, o_2^{source}, \cdots, o_n^{source}\}$ and $O_{stolen} = \{o_1^{stolen}, o_2^{stolen}, \cdots, o_n^{stolen}\}$ be the set of the outputs for the source and stolen models, where $o_k^{source}$ and $o_k^{stolen}$ represent the output of the source model and stolen model for the $k$-$th$ input sample. Then we can calculate the correlation matrix between all $n$ input samples and get a model-specific correlation matrix $C$ as follows:

$$C = \Phi(O), \quad C \in R^{n \times n} \quad where \quad C_{i,j} = corr(o_i, o_j) \quad i, j = 1, \cdots, n, \tag{1}$$

where $C_{i,j}$ represents the $i, j$ entry of the model's correlation matrix $C$, which can be calculated by the correlation between the $i$-$th$ and $j$-$th$ output of the model and $\Phi$ is the function of calculating the correlation matrix from the output set. To capture the complex correlations between the model's outputs, we introduce several functions to estimate the outputs' relationship[33]: First is cosine similarity [34]. Cosine similarity can be measured by the cosine of the angle between two vectors, expressed as:

$$C_{i,j} = Cos(o_i, o_j) = \frac{o_i^T o_j}{||o_i||||o_j||}, \quad i, j = 1, \cdots, n. \tag{2}$$

Another method for measuring outputs' correlation is Gaussian RBF [35]. Gaussian RBF is a popular kernel function, measuring two instances' distance based on their euclidean distances:

$$C_{i,j} = RBF(o_i, o_j) = exp(-\frac{||o_i - o_j||_2^2}{2\delta^2}), \quad i, j = 1, \cdots, n. \tag{3}$$

After calculating the correlation matrix for both the source model and the suspect model, we calculate the L1 distance between them and use it as our fingerprinting indicator. Any model's distance to the source model smaller than a threshold $d$ will be regarded as a stolen model:

$$Distance = \frac{||C_{stolen} - C_{source}||_1}{n^2} \leq d, \tag{4}$$

where $C_{stolen}$ and $C_{source}$ represent the correlation matrix of the stolen model and the source model. When the defenders want to determine the optimal threshold d in some situations, we can use the validation set to find the best threshold. For example, the defenders can use the average of the means of the correlation scores of the irrelevant models and the adversarial extracted models as d on the validation set.

Furthermore, there is a more strict situation in which the defender can only get access to the suspect model's predicted label but not the output probability. Facing this, we propose to use the predicted label to form a smooth-label probability. Smooth-label probability, as the soft form of the predicted label, has a similar form to the teacher's output and can replace the teacher's output in knowledge distillation [36]. In our fingerprinting method, we generate the smooth-label probability as follows:

$$p(x) = (1 - \epsilon)\delta(x) + \epsilon u(x), \quad \epsilon \in [0, 1], \tag{5}$$

where $p(x)$ is the smooth-label probability of the model input $x$, $\delta(x) = [0, 0, \cdots, 1, \cdots, 0]$ is the one-hot encoding for the predicted label, and u(x) is a uniform distribution. Similar to the model's output probability, the smooth label can perform well in our model fingerprinting SAC.

### 3.3 How to Find Suitable Samples?

Using the correlation matrix mentioned above, our fingerprinting method can robustly and efficiently identify model stealing attacks. Meanwhile, we also need suitable samples as model input to help fingerprint. All the previous fingerprinting methods use transferable adversarial examples to identify stolen models, but they are not robust to adversarial defense and can easily be cleared out by adversarial training. In fact, the attacker can evade detection by fine-tuning the extracted model only several epochs with same-distribution data and the source model's predicted label in an adversarial training way, expressed as follows:

$$\min_{\theta_{stolen}} \sum_i \max_{||\delta|| \leq \epsilon} l(f_{stolen}(x + \delta), f_{source}(x)), \tag{6}$$

where $f_{stolen}$ and $f_{source}$ represent the stolen and the source model, $\delta$ represents adversarial noise smaller than the bound $\epsilon$, which is generated by the stolen model using methods such as FGSM[37] or PGD[24] and $\theta_{stolen}$ represents the stolen model's parameters.

**Fingerprinting with Misclassified Normal Samples.** To solve the above problem, we propose to use the defender's training samples directly as the fingerprinting samples. Because there are hundreds or thousands of samples misclassified by the source model in the defender's dataset, we can use them as fingerprinting samples. The defender first uses the source model as the teacher to train several surrogate models by knowledge distillation [38]. Then the defender finds out the samples that both the source and the surrogate models predict wrongly and forms a hard sample dataset $D$:

$$D = [x_1, x_2, \cdots, x_n], \quad argmax(f_i(x_k)) \neq label_k \quad \forall i = 1, 2, \cdots, m, \tag{7}$$

where $x_1 \cdots x_n$ are all from the defender's dataset, $f_i$ represents the source and the surrogate models whose total number is $m$, and $label_k$ is the ground truth label for $x_k$. Furthermore, we can also make use of the irrelevant models trained with the defender's dataset, and choose the misclassified samples that are correctly classified by these irrelevant models to promote these misclassified samples' specificity only to the stolen models.

**Fingerprinting with Augmented Samples.** SAC-w can identify whether the suspect models are stolen from the source models, but it can be further improved. In some cases, the defender can not have enough normal samples to form the misclassified sample dataset. Furthermore, although SAC-w only needs to train much fewer surrogate models than CAE [7], it still consumes some computation resources of the defender. To distinguish the stolen models more efficiently and even more accurately, we propose to use CutMix [39] and image flip to augment data, and propose SAC-m, which uses these augmented samples as the model's input to calculate the models' sample correlation. CutMix generates a new training sample $(\tilde{x}, \tilde{y})$ using two training samples $(x_0, y_0)$ and $(x_1, y_1)$ by cutting off and pasting patches among them:

$$\tilde{x} = M \odot x_0 + (1 - M) \odot x_1, \quad \tilde{y} = \alpha y_0 + (1 - \alpha)y_1, \tag{8}$$

where $M \in \{0, 1\}^{W \times H}$ represents the binary mask to combine images in different part of images, $\odot$ represents an element-wise multiplication and $\alpha \in (0, 1)$ is a combination ratio [39]. In our fingerprinting method SAC-m, we can use CutMix once or several times to augment data. After CutMix, we can use image flip $x_{flip} = x^T$ on the mixed samples to augment data and reduce the accuracy further. With CutMix and image flipping, we form the CutMix Augmented samples. On one hand, CutMix increases the number of images, decreasing the number of normal samples needed by the defender. On the other hand, mixing up and image flipping generate harder samples. Because most models will agree on the same ground-truth label on normal samples, much harder augmented samples magnify the difference between two independent models and lower the influence of common knowledge across models. Besides, the correlation similarity between the source and the stolen model still holds up on these augmented samples. Thus, we can leverage these samples' correlation metric consistency and recognize the suspect models robustly and efficiently with only 100 or 200 normal samples from the defender's dataset.

## 4 Experiments

### 4.1 Setup

In this section, we evaluate different model intellectual property (IP) protection methods against different model stealing attacks on different datasets and model architectures, validating the effectiveness of our SAC-w and SAC-m methods.

We evaluate different IP protection methods against the following five categories of stealing attacks:

- **Fine-tuning.** In general, there are two common fine-tuning methods. One is fine-tuning the last layer (Finetune-L), which only fine-tunes the last layer and leaves the other layers unchanged. The other is fine-tuning all the layers (Finetune-A), which fine-tunes all the layers in the model. In our experiments, we assume the attacker fine-tunes the source model with an SGD optimizer using the attacker's dataset.
- **Pruning.** In our experiments, we leverage Fine Pruning [8] as our pruning methods. Fine Pruning, as a common backdoor defense method, prunes neurons according to their activation.
- **Transfer Learning.** The attacker may transfer the source model to another related task and make use of the source model's knowledge. We transfer both the CIFAR10 [40] model and the Tiny-ImageNet [41] model to other tasks and find out whether the fingerprinting methods can recognize. We transfer the CIFAR10 model to CIFAR10-C [42] and CIFAR100 dataset, of which we choose the front ten labels. Furthermore, we transfer Tiny-ImageNet model (trained with the front 100 labels in Tiny-ImageNet) to the 100 labels left behind in Tiny-ImageNet dataset.
- **Model Extraction.** Generally, there are two categories of model extraction, the probability-based model extraction, and the label-based model extraction. Label-based model extraction [10, 11] can only leverage the defender's predicted label to steal the knowledge of the source model and the loss function can be expressed as $L = CE(f_{stolen}(x), l_{source})$, where $l_{source}$ represents the label predicted by the source model and $CE(\cdot)$ represents the cross entropy loss. As for the probability-based model extraction [10, 43, 44], the attacker has access to the output probability and use it to train their stolen model:

$$L = \alpha \cdot KL(f_{stolen}^T(x), f_{source}^T(x)) + (1 - \alpha) \cdot CE(f_{stolen}(x), l_{source}), \quad (9)$$

where $f_{stolen}^T(x)$ and $f_{source}^T(x)$ represent the soft output of the stolen and source models, calculated by $f^T(x) = softmax(\frac{f(x)}{T})$, where $T$ represents the temperature, and $KL(\cdot)$ represents the KL divergence. In our experiments, we choose $T = 20$ as the temperature.

- **Adversarial Model Extraction.** Similar to the adaptive model extraction in CAE [7], the attacker can evade the fingerprinting detection by applying adversarial training after the label-based model extraction. But different from the adaptive model extraction, which uses the ground-truth label, we propose that the attacker can achieve the same goal with just the source models' predicted label in Equation 6. Thus, the attacker can leverage adversarial extraction with the same setting as simple model extraction and evade existing adversarial example based fingerprinting methods easily with a small accuracy decline.

**Model Architecture.**   We evaluate different IP protection methods on most of the common model architectures. All the extraction models and irrelevant models are trained on VGG [45], ResNet [46], DenseNet [47] and MobielNet [48]. Besides, in the transfer learning attack scenario, we use VGG architecture for both the transferred models and the irrelevant-new models. Specifically, we use VGG or ResNet model as the source model, and conduct experiments on different datasets across different architectures with different accuracy. Moreover, for every attack and irrelevant model architecture, we train 5 models each architecture to avoid contingency.

**Model IP Protection Methods.**   To validate our method's effectiveness, we compare it with several existing methods, including IPGuard [20], CAE [7] and EWE [5]. IPGuard and CAE leverage the transferability of adversarial examples and test these adversarial examples' attack success rate on the suspect models. Any model's attack success rate larger than a threshold will be recognized as a stolen model. EWE, on the contrary, trains the source model on backdoor data [49] and leaves the watermark in the model. Utilizing soft nearest neighbor loss to entangle the watermark data and the training data, EWE tries to increase the transferability of the watermark against model extraction.

Table 1: Different model intellectual protection methods distinguish irrelevant and stolen models on CIFAR10, where (−) represents that the protection method can not distinguish this kind of attack.

| Attack (AUC↑) | IPGuard [20] | CAE [7] | EWE [5] | SAC-w | SAC-m |
|---|---|---|---|---|---|
| Finetune-A | 1.00 | 1.00 | 1.00 | 1.00 | 1.00 |
| Finetune-L | 1.00 | 1.00 | 1.00 | 1.00 | 1.00 |
| Pruning | 1.00 | 0.95 | 0.87 | 1.00 | 1.00 |
| Extract-L | 0.81 | 0.83 | 0.97 | 1.00 | 1.00 |
| Extract-P | 0.81 | 0.90 | 0.97 | 1.00 | 1.00 |
| Extract-Adv | 0.54 | 0.52 | 0.91 | 1.00 | 0.92 |
| **Average** | 0.86 | 0.87 | 0.95 | 1.00 | 0.99 |
| Transfer-10C | 1.00 | 1.00 | 1.00 | 1.00 | 1.00 |
| Transfer-A | − | − | − | 1.00 | 1.00 |
| Transfer-L | − | − | − | 1.00 | 1.00 |

**Datasets.** To validate different fingerprinting methods' effectiveness and robustness, we conduct experiments on different datasets. Same as previous works, we split the training dataset into two same-size parts $D_{defender}$ and $D_{attacker}$, which the defender and the attacker own respectively. Because the number of samples for one label in Tiny-ImageNet is only 250 and the source model accuracy will decrease to about $40\%$, we choose the front 100 labels to form a smaller dataset to achieve better source model accuracy.

**Evaluation Metrics.** To evaluate the effectiveness of different fingerprinting methods, same as CAE[7], we leverage AUC-ROC curve [50] and use AUC value between the fingerprinting scores of the irrelevant models and the stolen models to measure the fingerprinting effectiveness. ROC is a probability curve, which is plotted with True Positive Rate and False Positive Rate. AUC represents the area under ROC curve (max is 1) and a larger AUC represents a better fingerprinting method. Besides, when $AUC = 0.5$, the fingerprinting detection can be viewed as a random guess. Furthermore, to better depict the influence of model architectures and initialization on fingerprinting, we introduce the fingerprinting score (Score) to represent the indicator that different methods use to distinguish the suspect models. To be specific, the Score represents the attack success rate in IPGuard, CAE, and EWE, and represents the correlation difference between the suspect model and the source model in SAC-w and SAC-m.

Table 2: Different model intellectual protection methods distinguish irrelevant and stolen models on Tiny-ImageNet, where (−) represents that the protection method can not distinguish this kind of attack.

| Attack(AUC↑) | IPGuard [20] | CAE [7] | EWE [5] | SAC-w | SAC-m |
|---|---|---|---|---|---|
| Finetune-A | 1.00 | 1.00 | 0.48 | 1.00 | 1.00 |
| Finetune-L | 1.00 | 1.00 | 1.00 | 1.00 | 1.00 |
| Pruning | 1.00 | 1.00 | 0.58 | 1.00 | 1.00 |
| Extract-L | 0.97 | 1.00 | 1.00 | 1.00 | 1.00 |
| Extract-P | 0.97 | 1.00 | 1.00 | 1.00 | 1.00 |
| Extract-Adv | 0.65 | 0.78 | 1.00 | 0.92 | 0.91 |
| **Average** | 0.93 | 0.96 | 0.84 | 0.99 | 0.99 |
| Transfer-A | − | − | − | 1.00 | 1.00 |
| Transfer-L | − | − | − | 1.00 | 1.00 |

Table 3: Accuracies (%) on CIFAR10 of the source model and different attack models.

| % | Irrelevant | | | | Finetune | | Pruning | | CIFAR10-C | | CIFAR100 | | |
|---|---|---|---|---|---|---|---|---|---|---|---|---|---|
| Source | VGG | ResNet | Dense | Mobile | F-A | F-L | p=0.3 | p=0.5 | Irrelevant | FT | Irrelevant | T-A | T-L |
| 87.3 | 87.8 | 82.1 | 81.0 | 77.0 | 88.1 | 87.4 | 84.7 | 83.3 | 76.5 | 77.0 | 72.4 | 77.9 | 34.0 |
| | Extract-L | | | | Extract-P | | | | Extract-Adv | | | | |
| VGG | ResNet | Dense | Mobile | VGG | ResNet | Dense | Mobile | VGG | ResNet | Dense | Mobile | |
| 86.2 | 82.7 | 83.0 | 80.9 | 86.8 | 84.1 | 85.0 | 83.8 | 83.6 | 80.2 | 79.4 | 77.1 | |

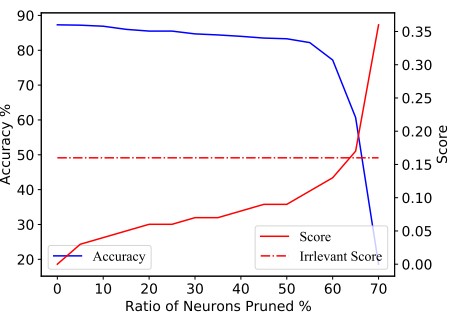 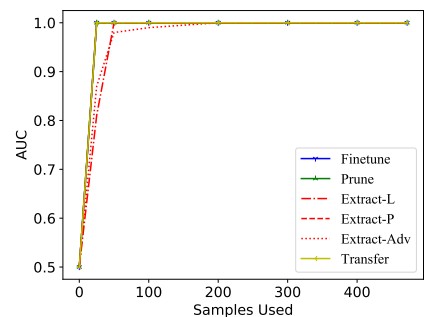

(a) SAC-w's score and model's accuracy change during pruning.

(b) SAC-w's AUC change with different amounts of misclassified samples.

Figure 2: SAC's performance change with different pruning ratio and different sample amounts.

## 4.2 Fingerprinting against Different Attacks

Tables 1 and 2 demonstrate different fingerprinting methods against different model stealing attacks on CIFAR10 and Tiny-ImageNet. **Finetune-A** and **Finetune-L** represent fine-tuning the source model on all the layers and the last layer, and **Extract-L**, **Extract-P** and **Extract-Adv** represent the three settings of the model extraction, label-based model extraction, probability-based model extraction and adversarial model extraction. Besides, **Transfer-A** and **Transfer-L** represent transferring the source model to a new dataset by fine-tuning all the layers or the last layer, and **Transfer-10C** represents transferring the source model to CIFAR10C. Moreover, for better comparison, we calculate the average AUC of different model intellectual protection methods facing different model stealing attacks. Specifically, **Average** represents the average AUC of fine-tuning, pruning and model extraction, and **Transfer-Average** represents the average AUC of different transferring attacks. Tables 1 and 2,illustrate the effectiveness and robustness of our SAC-w and SAC-m. In most situations, our method has a better AUC than other model IP protection methods. Furthermore, to demonstrate our SAC-m method's effectiveness across different datasets or architectures, we conduct more experiments, which are shown in Appendix. From the experiments, the average attack success rate for CAE is larger than IPGuard, indicating CAE has better transferability than IPGuard. Besides, we find CAE is better than IPGuard in identifying the model extraction attacks, which is mainly because of the introduction of conferrable scores. But these methods' attack success rate still fluctuates largely across different model architectures. On the contrary, our method SAC has a better AUC than the other two fingerprinting methods. Furthermore, our method SAC-m does not need to train any surrogate models for the defender, which saves the defender's time largely. Besides, on Tiny-ImageNet, our experiments illustrate that SAC-m can fingerprint accurately with only CutMix samples and we use these samples directly without flipping on Tiny-ImageNet. When facing adversarial extraction, adversarial example based method IPGuard and CAE's AUC declines sharply to near $0.5$, which is also illustrated in [7], and $AUC = 0.5$ represents a random guess. Furthermore, our experiments demonstrate that EWE is sensitive to pruning, although its AUC decline is smaller than other normal backdoor-based watermark methods. On the contrary, our two methods SAC-w and SAC-m have the best detection performance when facing most model stealing attacks. Furthermore, the Score of the stolen models used by our methods has a much smaller variance than the other three methods. Because the attack success rate of the adversarial examples or model watermark will be influenced by the robustness of the model, models with a more robust architecture or after adversarial training, which even is stolen from the source model, may have a lower attack success rate than the irrelevant models. On the contrary, correlation difference is not related to the robustness of the model architectures and can better detect the stolen model across different model architectures. Furthermore, because all the other three methods use the attack success rate as the only identification criteria, when facing transfer learning or label changing, they are not able to identify the stolen models. On the contrary, our methods use the correlation difference as the fingerprint and the correlation consistency holds up when the models are transferred to another task.

Table 3 illustrates the average accuracy of the source and stolen models, demonstrating that most of the model stealing attacks are able to steal the source model with only a small accuracy decline.

Table 4: Correlation fingerprinting with different attacker's accessibility.

| AUC↑ | FT-A | FT-L | Pruning | Ex-L | Ex-P | Ex-Adv | Trans-10C | Trans-A | Trans-L |
|---|---|---|---|---|---|---|---|---|---|
| Smooth-label | 1.0 | 1.0 | 1.0 | 1.0 | 1.0 | 1.0 | 1.0 | 0.0 | 0.9 |
| Probability | 1.0 | 1.0 | 1.0 | 1.0 | 1.0 | 1.0 | 1.0 | 1.0 | 1.0 |

Table 5: Time consumption and source model's accuracy decline on CIFAR10.

| | IPGuard [20] | CAE [7] | EWE [5] | SAC-w | SAC-m |
|---|---|---|---|---|---|
| Time | $456.45s$ | $25,536.89s$ | $277.31s$ | $2,380.19s$ | $4.45s$ |
| Accuracy (%) | 87.3 | 87.3 | $83.3(-4.0)$ | 87.3 | 87.3 |

Furthermore, we also take the situation that the defender can only access the defender's predicted label into account. Table 4 demonstrates that even only with the predicted label, our correlation-based method recognizes the stolen models effectively and robustly, where smooth-label represents the situation that only the predicted label is available and probability represents the output probability is available. **FT**, **Ex**, and **Trans** are abbreviation for **Finetune**, **Extract** and **Transfer**. Only with the predicted label, SAC can also succeed in two hard tasks, adversarial extraction, and Transfer-L. But SAC fails in Transfer-A when there is only the hard label available, and it is because of the bigger changes in Transfer-A models than Transfer-L models. Compared with Transfer-L, the feature layers in the Transfer-A model have a much larger difference from the source model, which makes it more difficult for the defenders to detect Transfer-A than Transfer-L. Figure 2(a) demonstrates the source model's accuracy and SAC-w's scores' change during pruning the neurons on CIFAR10. In the figure, Score represents the score of the source model after pruning the neurons, and Irrelevant Score represents the average score of the irrelevant models with different model architectures. With the pruning ratio increasing, the source model's accuracy declines, and the score for SAC-m increases. We find that only when the pruning ratio is over $60\%$, SAC-w can not identify the stolen models. However, when the pruning ratio is that high, the stolen model's accuracy is largely lower than the irrelevant models, and the pruned model with that low accuracy can not be used, indicating our SAC's effectiveness.

## 4.3 Influence of Fingerprinting on the Defender

When the defender wants to fingerprint or watermark a model, there will always be some sacrifice, such as a larger time consumption or some accuracy decline in the source models. Table 5 demonstrates different IP protection methods' time consumption and influence on the source model. First, we focus on the accuracy decline, and only EWE causes the source model's accuracy to decline. Because model fingerprinting methods, including IPGuard, CAE, SAC-w, and SAC-m, do not involve in the source model's training procedure, there is no accuracy loss. On the contrary, EWE needs to train the source model on the watermark data, leading to an accuracy decline, i.e. $4.0\%$ accuracy decline on CIFAR10. As for the time consumption, CAE consumes several hours to train surrogate models and form adversarial examples, causing a great computation burden for the defenders. Moreover, on a larger dataset such as ImageNet, CAE may even take up more time to form the fingerprint. On the contrary, our SAC-m method costs only 4.45 seconds to form our fingerprint, which is 5,738 faster than the CAE method.

Table 6: Ablation study on CIFAR10.

| Attack(AUC↑) | SAC-m | SAC-normal | SAC-source |
|---|---|---|---|
| Finetune-A | 1.00 | 1.00 | 1.00 |
| Finetune-L | 1.00 | 1.00 | 1.00 |
| Pruning | 1.00 | 1.00 | 1.00 |
| Extract-L | 1.00 | 0.88 | 0.88 |
| Extract-P | 1.00 | 1.00 | 1.00 |
| Extract-Adv | 0.92 | 0.78 | 0.75 |
| **Average** | 0.99 | 0.94 | 0.94 |
| Transfer-10C | 1.00 | 0.99 | 0.99 |
| Transfer-A | 1.00 | 1.00 | 1.00 |
| Transfer-L | 1.00 | 1.00 | 1.00 |
| **Transfer-Average** | 1.00 | 1.00 | 1.00 |

Table 7: Correlation function's influence on model fingerprinting on CIFAR10.

| AUC↑ | FT-A | FT-L | Pruning | Ex-L | Ex-P | Ex-Adv | Trans-10C | Trans-A | Trans-L |
|---|---|---|---|---|---|---|---|---|---|
| Gaussian | 1.0 | 1.0 | 1.0 | 1.0 | 1.0 | 1.0 | 1.0 | 1.0 | 1.0 |
| Cosine | 1.0 | 1.0 | 1.0 | 1.0 | 1.0 | 1.0 | 1.0 | 1.0 | 1.0 |

## 4.4 Ablation Study

In this subsection, we will take the input samples' choice, correlation's choice, and correlation function into consideration to validate our SAC-w and SAC-m's effectiveness. In Table 6, SAC-normal represents fingerprinting correlation with normal samples from the defender's dataset and SAC-source represents fingerprinting correlation with source model misclassified samples. Table 6 illustrates SAC-m is only more effective than fingerprinting with normal samples or source model's misclassified samples, but also declines the need for samples from the defender's dataset, where SAC-m, SAC-normal, and SAC-source both use 100 normal samples in the experiment. Because of the lack of normal samples for SAC-source, its performance is affected. Our experiments demonstrate that only with 50 or 100 normal samples, SAC-m fingerprints the source model accurately on CIFAR10. On the contrary, SAC-w usually needs thousands of normal samples. Moreover, we also consider the different correlation function's influence on the model fingerprinting. Table 7 illustrates that both the Gaussian RBF kernel and cosine similarity fingerprint the source model successfully. Because the correlation using cosine similarity has a larger gap between the irrelevant models and the stolen models, we choose the cosine similarity as our correlation function in the above experiments. Furthermore, to verify SAC's effectiveness with different numbers of misclassified samples, we conduct experiments with SAC-w on CIFAR10, and the results are shown in Figure 2(b). The above experiment result demonstrates that our SAC method is robust even under the few-shot setting. And only with 50 misclassified samples, SAC can successfully detect different kinds of model stealing attacks.

## 5 Conclusion and Limitation

In this paper, we propose a correlation-based fingerprinting framework to detect different model stealing attacks. Different from previous adversarial example based fingerprinting methods, our methods SAC-w and SAC-m make use of misclassified normal samples or CutMix Augmented samples as the model input and calculate the correlation difference as the indicator to recognize the stole models without involving in the model's training procedure. SAC achieves the best model stealing attack detection performance, with $AUC = 1$ in most cases, across different model architectures and datasets. Furthermore, SAC is able to identify the model stealing attacks with adversarial training or transfer learning techniques, which can not be detected by previous fingerprinting methods. Moreover, SAC-m does not need to train the surrogate models or generate adversarial examples, and thus, is much faster than other fingerprinting methods (5,738 faster than CAE). Although SAC-m is robust to adversarial extraction (training), SAC-m still performs a little worse facing adversarial extraction and we leave finding for a better data augmentation method to future work. Moreover, selecting an optimal $d$ for the model stealing detection is also important, we will also leave it to future work.

## 6 Broader Impact

Our correlation-based model fingerprinting method, as a model IP protection method, will help both academia and industry to protect their model's intellectual property and prevent from model's illegal usage. However, on the other hand, the protection of the model owner's rights may promote the development of machine learning as a service, and potentially bring social unemployment.

## 7 Acknowledgement

This work is partially funded by National Natural Science Foundation of China (Grant No. U21B2045, U20A20223) and Beijing Nova Program under Grant Z211100002121108.

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
