# —Appendix—
# Are You Stealing My Model? Sample Correlation for Fingerprinting Deep Neural Networks

**Jiyang Guan**[1,2], **Jian Liang**[1,2], **Ran He**[1,2*]

[1]NLPR &CRIPAC, Institute of Automation, Chinese Academy of Sciences, China
[2]School of Artificial Intelligence, University of Chinese Academy of Sciences, China
`guanjiyang2020@ia.ac.cn, liangjian92@gmail.com, rhe@nlpr.ia.ac.cn`

## 1 Setup

The experiments are conducted on a Linux server equipped with an Intel(R) Xeon(R) Gold 6130 CPU @ 2.10GHz, 256GB RAM, and 8 NVIDIA RTX 3090 GPUs (with 24GB memory each). All models are implemented in PyTorch version 1.11.0 with CUDA version 11.3, and Python 3.8.

To verify our fingerprinting method's effectiveness, we conduct experiments on FashionMNIST [1], CIFAR10 [2], CIFAR100 [2], and Tiny-ImageNet [3] four datasets across VGG, ResNet, MobileNet, and DenseNet four common model architectures. All the datasets used in this paper are open-sourced. The detailed information about the datasets used in our experiments is illustrated in Table 1.

Table 1: Detailed information about datasets

| Dataset | labels | Image size | Training Images |
|---------|--------|------------|-----------------|
| FashionMNIST | 10 | $28 \times 28 \times 1$ | 60,000 |
| CIFAR10 | 10 | $32 \times 32 \times 3$ | 60,000 |
| CIFAR100 | 100 | $32 \times 32 \times 3$ | 60,000 |
| Tiny-ImageNet | 200 | $64 \times 64 \times 3$ | 100,000 |

The licenses for the datasets used in this paper are as follows: License for CIFAR10 is `https://github.com/wichtounet/cifar-10/blob/master/LICENSE`. License for CIFAR100 is `https://github.com/JinLi711/CIFAR-100/blob/master/LICENSE`. License for Tiny-ImageNet is `https://github.com/DennisHanyuanXu/Tiny-ImageNet/blob/master/LICENSE`. License for FashionMNIST is `https://github.com/zalandoresearch/fashion-mnist/blob/master/LICENSE`.

Table 2: Accuracies (%) on CIFAR10 under different source models and attacks for EWE [4]

| % | Irrelevant | | | | Finetune | | Pruning | | CIFAR10-C | | |
|---|-----|--------|-------|--------|-----|-----|-------|-------|------------|-----|---|
| Source | VGG | ResNet | Dense | Mobile | F-A | F-L | p=0.2 | p=0.3 | Irrelevant | FT | |
| 83.3 | 87.8 | 82.1 | 81.0 | 77.0 | 89.4 | 85.3 | 85.4 | 85.1 | 76.5 | 77.0 | |
| Extract-L | | | | Extract-P | | | | Extract-Adv | | | |
| VGG | ResNet | Dense | Mobile | VGG | ResNet | Dense | Mobile | VGG | ResNet | Dense | Mobile |
| 83.5 | 78.5 | 77.5 | 73.2 | 82.4 | 78.2 | 76.8 | 72.9 | 77.7 | 72.2 | 70.7 | 65.4 |

Furthermore, because EWE needs to involve in the model's training procedure, the source model for EWE is different from the source model used in the other three fingerprinting methods. Thus, all the stealing attack models are retrained and the average accuracy of the source model and the attack models is demonstrated in Table 2 for the EWE [4]. In the table, the p in Pruning represents

---

[*]Corresponding Author

the pruning ratio, FT in CIFAR10-C represents transferring the source model to CIFAR10-C. Besides, VGG, ResNet, Dense, and Mobile represent VGG model, ResNet model, DenseNet model, and MobileNet model respectively. Because EWE uses the watermarking method and involves in the model training procedure, there is a $4\%$ accuracy drop of the source model compared with the original model, $83.3\%$ of EWE's source model vs $87.3\%$ of the original source model. Furthermore, after fine-tuning either all layers or last layer, the accuracy of EWE's stolen models is higher than the EWE's source model, indicating that EWE disturbs the model's training procedure largely. EWE makes a trade-off between the robustness of the watermark against model extraction and the source model's accuracy, which will cause harm to the source model's training and utilization.

## 2 Fingerprinting on Different Datasets

Table 3: Different model intellectual protection methods distinguish irrelevant and stolen models on FashionMNIST.

| Attack | IPGuard [5] | CAE [6] | SAC-m |
|---|---|---|---|
| Finetune-A | 1.00 | 1.00 | 1.00 |
| Finetune-L | 1.00 | 1.00 | 1.00 |
| Pruning | 1.00 | 0.99 | 1.00 |
| Extract-L | 0.62 | 0.73 | 0.95 |
| Extract-P | 0.57 | 0.80 | 0.99 |
| Extract-Adv | 0.34 | 0.36 | 0.89 |
| **Average** | 0.76 | 0.81 | 0.97 |

Table 4: Different model intellectual protection methods distinguish irrelevant and stolen models on CIFAR100.

| Attack | IPGuard [5] | CAE [6] | SAC-m |
|---|---|---|---|
| Finetune-A | 1.00 | 1.00 | 1.00 |
| Finetune-L | 1.00 | 1.00 | 1.00 |
| Pruning | 1.00 | 1.00 | 1.00 |
| Extract-L | 0.70 | 0.79 | 0.95 |
| Extract-P | 0.80 | 0.94 | 1.00 |
| Extract-Adv | 0.58 | 0.65 | 0.91 |
| **Average** | 0.85 | 0.90 | 0.98 |

To demonstrate the effectiveness of SAC-m across different datasets further, we evaluate on two more datasets FashionMNIST and CIFAR100. Tables 3 and 4 demonstrate the results of IPGuard, CAE, and SAC-m on the FashionMNIST and CIFAR100 datasets. From the experiments, the average attack success rate for CAE is larger than IPGuard, indicating CAE has a better transferability than IPGuard. Besides, we find CAE is much better than IPGuard in identifying the model extraction attacks, which is mainly because of the introduction of conferrable scores. But these methods' attack success rate still fluctuates largely across different model architectures. On the contrary, our method SAC has a better AUC than the other two fingerprinting methods. Furthermore, our method does not need to train any surrogate models for the defender, which saves the defender's time largely.

## 3 Fingerprinting with Different Source Model Architectures

Because the above experiments are based on a VGG source model condition, we try to find out whether SAC fingerprints the source model successfully on different model architectures. In Table 5, we do experiments with a ResNet-based source model and test different fingerprinting methods' effectiveness against different model stealing attacks. Our experiments demonstrate that although facing a different source model architecture, SAC can still identify the stolen model successfully, including fine-tuning, pruning, model extraction, adversarial model extraction and transfer learning. Besides, Transfer-A and Transfer-L in Table 5 represents transferring the source model on CIFAR10 to the front 10 labels on CIFAR100.

Table 5: Different model intellectual protection methods distinguish irrelevant and stolen models on CIFAR10 with ResNet34 souce model, where $(-)$ represents that the protection method can not distinguish this kind of attack.

| Attack | IPGuard [5] | CAE [6] | SAC-m |
|---|---|---|---|
| Finetune-A | 1.00 | 1.00 | 1.00 |
| Finetune-L | 1.00 | 1.00 | 1.00 |
| Pruning | 1.00 | 1.00 | 1.00 |
| Extract-L | 0.61 | 0.81 | 1.00 |
| Extract-P | 0.66 | 0.92 | 1.00 |
| Extract-Adv | 0.54 | 0.63 | 0.94 |
| **Average** | 0.80 | 0.89 | 0.99 |
| Transfer-A | − | − | 1.00 |
| Transfer-L | − | − | 1.00 |