# OpenReview forum: "Are You Stealing My Model? Sample Correlation for Fingerprinting Deep Neural Networks"
_NeurIPS.cc/2022/Conference — NeurIPS 2022 Accept_

### Official Review · Reviewer_LN2K · 2022-07-09

**Rating:** 6
**Confidence:** 4
**Soundness:** 2 fair
**Presentation:** 2 fair
**Contribution:** 2 fair

**Summary:**

This paper proposes a model stealing detection method based on sample correlation. The proposed method calculated the correlation among the model outputs for the misclassified samples. The cutmix approach is used to generate more effective sample inputs. The experiment evaluates the performance of the proposed method outperforms in different attack scenarios, such as fine-tuning, transfer learning, and adversarial training.

**Questions:**

1. What is the threat model in each attack category?

2. Why the misclassified samples can help to identify the stolen model?

**Limitations:**

The authors have discussed potential societal impact of their work.

**Strengths And Weaknesses:**

Strengths:

1. The paper considers many realistic attack scenarios in model stealing attacks, such as fine-tuning, pruning, adversarial training, and transfer learning, which have not been widely explored.

2. It is nice to see that the protection method considers the label-only cases.

3. Leveraging CutMix to augment data for fingerprinting is novel.


Weaknesses:

1. The threat model is not well-defined in the paper. What are the attacker’s capabilities in each category of stealing attacks? Does the attacker have access to training data and model architectures?

2. The implementation of the model stealing attacks is unclear.

3. It is unclear why the misclassified samples help to identify the stolen model. How many misclassified samples are necessary for fingerprinting?

4. The threshold d in Equation (4) is critical to the success of model stealing detection. However, it is unclear how to select an appropriate value of d.

5. As shown in Table 1 and 2, the existing detection methods can achieve 100% AUC in many attack categories (e.g., fine-tune, pruning). The proposed method only outperforms the existing methods in model extraction attacks.

6. The proposed method does not work well in the Transfer-A attack (Table 4). It would be great if the paper could provide some explanations on it.

---

> ### Author Response · Authors · 2022-08-02
> **Response to Reviewer LN2K (Part 1)**
>
> Thank you for the detailed reviews, as well as the suggestions for improvement. And we hope to resolve some of your concerns in the following comments.
>
> **Q1.**
> The threat model is not well-defined in the paper. What are the attacker's capabilities in each category of stealing attacks? Does the attacker have access to training data and model architectures?
>
> **A1.**
> We share the same setting as most fingerprinting methods, e.g. CAE [6, ICLR2021]. The attacker and the defender hold different datasets, the attacker's dataset, and the defender's dataset. In our experiments, we divide the training set of CIFAR10, CIFAR100, FashionMNIST, and Tiny-ImageNet into two equal pieces as the attacker's dataset and the defender's dataset. Furthermore, we list five model stealing attacks in our paper, fine-tuning, pruning, transfer learning, model extraction, and adversarial model extraction. Attackers for fine-tuning, pruning, transfer learning have access to their own datasets (the attacker's dataset) and the source model (including the model architecture and the inner parameters). On the other hand, attackers for model extraction and adversarial model extraction do not have access to the source model inner parameters or the source model architecture. They only use their own unlabeled dataset (the attacker's dataset without labels) and the output from the source model to train their own models.
>
>
> **Q2.**
> The implementation of the model stealing attacks is unclear. What is the threat model in each attack category?
>
> **A2.**
> In line 208 to line 235 in this paper, we list all the implementation details about model stealing attacks. In our paper, all the IP protection methods are conducted experiments based on the same attacker's setting and try to detect these model stealing attacks. The different model stealing attacks listed in our paper are based on the setting of the former papers CAE [6, ICLR2021], IPGuard [19, ICCCS2021], and EWE [4, USENIX2021]. We list five model stealing attacks in our paper and show their implementation below:
> * **Fine-tuning:** The attackers can have access to the source model (including the source model's inner parameters) and try to avoid the model owner's detection by fine-tuning the model on their own datasets, the attacker's dataset. In our experiments, we assume the attacker fine-tunes the source model for 30 epochs with SGD optimizer with lr=5e-4 in either all the layers (Finetune-A) or the last layer (Finetune-L).
> * **Pruning:** The attackers can have access to the source model (including the source model's inner parameters) and try to avoid the model owner's detection by pruning the model on their own datasets, the attacker's dataset. The attackers in our experiments use Fine Pruning to avoid detection. Fine Pruning prunes the neurons in the order according to their activation and fine-tunes the model each time after pruning 50 neurons to maintain the model's accuracy.
> * **Transfer Learning:** The attackers can have access to the source model (including the source model's inner parameters) and a new dataset to which they want to transfer the source model. Then they train the well-trained source model to other datasets, e.g. CIFAR10 model to CIFAR10C (we choose snow as the corruption) or CIFAR100. We assume that the attacker retrains the source model on the new dataset with all images from the attacker's dataset for 30 Epoches with lr=5e-4.
> * **Model Extraction and Adversarial Model Extraction:** There are three common model extraction methods listed in our paper, label-based model extraction, probability-based model extraction, and adversarial model extraction. During model extraction, the attacker can not have access to the source model's inner parameters, and what they can have access to is the unlabeled dataset (attacker's dataset, but without labels) and the source models' output on these unlabeled data. Label-based model extraction can only have access to the source model's output labels and use them with their unlabeled data to train their models. Probability-based model extraction can have access to the source model's output probability and use Equation 9 in our paper to train their own models. Furthermore, based on label-based extracted models, the attacker can further avoid adversarial-example based model fingerprinting by adversarial training on the extracted models, with Equation 6 in our paper. Various model architectures are verified for the model extraction methods, including VGG, ResNet, DenseNet, and MobileNet to test different model IP protection models' effectiveness and robustness.

---

> > ### Author Response · Authors · 2022-08-02
> > **Response to Reviewer LN2K (Part 2)**
> >
> > **Q3.**
> > It is unclear why the misclassified samples help to identify the stolen model. How many misclassified samples are necessary for fingerprinting?
> >
> >  **A3.**
> > As we stated around line 57, correctly classified samples hold common knowledge which most models will hold, and this will cause both the irrelevant models and the stealing models to produce similar right output on these samples, which will cause a problem to distinguish the stealing models and the irrelevant models. Furthermore, this phenomenon is also found in adversarial example based fingerprinting methods, such as CAE [6, ICLR2021], and IPGuard [19, ICCCS2021]. In our experiments, we use 472 misclassified samples in SAC-w and 50 normal samples to do data augmentation in SAC-m in our CIFAR10 experiments in Table 1 in our paper. Furthermore, to verify SAC's effectiveness with different numbers of samples, we conduct experiments with SAC-w on CIFAR10 (the same setting as Table 1 in our paper), and the results are as follows:
> >
> > | Samples Used |Finetune-A | Finetune-L| Pruning |Extract-L | Extract-P | Extract-Adv| Finetune-10C |Transfer-A |Transfer-L |
> > | ------- | ------- |------- | -------  |------- | ------- |------- | -------  |-------  |-------  |
> > | 472 |1.00 | 1.00| 1.00 |1.00 |1.00 | 1.00| 1.00 |1.00 |1.00 |
> > | 300 |1.00 | 1.00| 1.00 |1.00 |1.00 | 1.00| 1.00 |1.00 |1.00|
> > | 200 |1.00 | 1.00| 1.00 |1.00 |1.00 | 1.00| 1.00 |1.00 |1.00 |
> > | 100 |1.00 | 1.00| 1.00 |1.00 |1.00 | 0.99| 1.00 |1.00 |1.00 |
> > | 50 |1.00 | 1.00| 1.00 |0.99 |1.00 | 0.98| 1.00 |1.00 |1.00 |
> > | 25 |1.00 | 1.00| 1.00 |0.81 |1.00 | 0.88| 1.00 |1.00 |1.00 |
> >
> > The above experiment result demonstrates that our SAC method is robust even under the few-shot setting. And only with 50 samples, SAC can successfully detect different kinds of model stealing attacks.
> >
> > **Q4.**
> > The threshold d in Equation (4) is critical to the success of model stealing detection. However, it is unclear how to select an appropriate value of d.
> >
> >  **A4.**
> > To evaluate the performance of detecting model stealing attacks, AUC (Area under the ROC Curve) is the most widely-used evaluation metric in this field, which does not rely on the threshold value d.
> > For example, both CAE [6, ICLR2021] and IPGuard [19, ICCCS2021] employ AUC to measure the detection performance of different stealing model detection methods.
> > When the defenders want to determine the optimal threshold d in some situations, we can use the validation set to find the best threshold. For example, the defenders can use the average of the means of the correlation scores of the irrelevant models and the adversarial extracted models as d on the validation set. We will also leave it for our future work.
> >
> > **Q5.**
> > As shown in Tables 1 and 2, the existing detection methods can achieve 100% AUC in many attack categories (e.g., fine-tune, pruning). The proposed method only outperforms the existing methods in model extraction attacks.
> >
> > **A5.**
> > The different model stealing attacks listed in our paper are based on the setting of the former papers CAE [6, ICLR2021], IPGuard [19, ICCCS2021], and EWE [4, USENIX2021]. Thus, there are some model stealing attacks which can be easily detected by all the model IP protection methods, such as fine-tuning. Our SAC method outperforms former model IP protection methods such as CAE [6, ICLR2021], IPGuard [19, ICCCS2021], and EWE [4, USENIX2021], on both different kinds of model extraction attacks and transfer learning attacks. All the three compared methods CAE, IPGuard, and EWE fail when there is a transfer learning attack because of the label space change after transfer learning. Furthermore, because the defenders do not know exactly how the attackers steal their models, such as transfer learning or model extraction, the attacker can evade the defender's detection with these methods, causing a great threat. Our method, on the contrary, can detect all these model stealing attacks with a high AUC. Furthermore, our method does not need to be involved in the training process like EWE, and thus our method will not harm the source model's accuracy.

---

> > > ### Author Response · Authors · 2022-08-02
> > > **Response to Reviewer LN2K (Part 3)**
> > >
> > > **Q6.**
> > > The proposed method does not work well in the Transfer-A attack (Table 4). It would be great if the paper could provide some explanations on it.
> > >
> > > **A6.**
> > > Transfer learning (A) represents training the source model in all the layers on a new dataset. Compared with transfer learning in the last layers (transfer learning (L)), the feature layers in the transfer learning (A) model have a much larger difference from the source model, which makes it more difficult for the defenders to detect transfer learning (A) than transfer learning (L). Furthermore, because SAC-w and SAC-m use the images from the original datasets as the model's inputs, the transferred model on the new datasets generates a more soft and uncertain output. Thus, the transferred model may output [0.1, 0.1, 0.4, 0.3] and [0.1, 0.1, 0.3, 0.4] on two images belonging to the same class in the source model datasets. Although their output probabilities are similar, indicating they are more likely to belong to the same class, their outputs' hard labels are different, and it may cause SAC to fail under hard label setting with transfer learning (A). We have also explained it in the **revised paper**.

---

> ### Author Response · Authors · 2022-08-08
> **Thank you again for your thoughtful review. Does our response address your questions? We would appreciate the opportunity to engage further if needed.**
>
> Dear reviewer,
>
> Thank you again for your thoughtful review. Does our response address your questions? We would appreciate the opportunity to engage further if needed.

---

> > ### Comment · Reviewer_LN2K · 2022-08-09
> > **Thank you for the response**
> >
> > The response addresses my questions. I hope the clarification and new results will be included in the paper revision.

---

> > > ### Author Response · Authors · 2022-08-09
> > > **Thanks for your support! We will include the clarification and new results.**
> > >
> > > Thanks for your support! We will include the clarification and new results.

---

### Official Review · Reviewer_4uwd · 2022-07-09

**Rating:** 7
**Confidence:** 4
**Soundness:** 3 good
**Presentation:** 3 good
**Contribution:** 3 good

**Summary:**

Model fingerprinting allows a model owner to claim ownership of a stolen model. Prior works on fingerprinting typically use transferable adversarial examples to perform fingerprinting. Such techniques have two key shortcomings: 1. They don’t work in the presence of defenses like adversarial training 2. They cannot be used when the stolen model is used for transfer learning as the output label space is different from the original model. To solve these issues, the authors propose to use pair-wise correlation of the model’s output of wrongly classified (SAC-w) or mixed (SAC-m) samples to perform fingerprinting.  Using wrongly classified/mixed inputs (instead of adversarial examples) allows the technique to be used in the presence of defenses like adversarial training. Using pair-wise relationships (instead of point-wise predictions) allows fingerprinting to be performed when the stolen model is used for transfer learning. Evaluations show the the proposed technique can detect stolen models with high accuracy and can outperform prior works.

**Questions:**

1.  Is there a specific reason for why SAC fails under hard label setting with transfer learning?
2. There is a recent work that proposes a similar solution of measuring pairwise responses to fingerprint models: https://arxiv.org/pdf/2106.12478.pdf. It would be good to cite this paper and point out differences compared to the proposed approach.

**Limitations:**

The authors have addressed the limitations and societal impact of their work.

**Strengths And Weaknesses:**

## Strengths

1. The paper proposes a technique to perform fingerprinting in the presence of transfer learning, which is a new and interesting sub-problem in model fingerprinting that has received limited attention from prior works.
2. In addition to the soft-label setting, the paper proposes a method to convert hard labels to soft labels, which enables fingerprinting when only the hard labels are available.
3. The proposed fingerprinting techniques have been evaluated against several categories of stealing attacks: fine-tuning, pruning, transfer learning, model extraction and adversarial model extraction. The proposed techniques shows high detection performance for fingerprinting and outperforms prior works.
4. The paper is well-written and easy to follow.

## Weakness

1. SAC does not seem to work under transfer learning (A) in the hard label setting (Table 4).

---

> ### Author Response · Authors · 2022-08-02
> **Response to Reviewer 4uwd**
>
> Thank you for the detailed reviews, as well as the suggestions for improvement. And we hope to resolve some of your concerns in the following comments.
>
> **Q1.**
> SAC does not seem to work under transfer learning (A) in the hard label setting (Table 4). Is there a specific reason for why SAC fails under hard label setting with transfer learning?
>
> **A1.**
> Transfer learning (A) represents training the source model in all the layers on a new dataset. Compared with transfer learning in the last layers (transfer learning (L)), the feature layers in the transfer learning (A) model have a much larger difference from the source model, which makes it more difficult for the defenders to detect transfer learning (A) than transfer learning (L). Furthermore, because SAC-w and SAC-m use the images from the original datasets as the model's inputs, the transferred model on the new datasets generates a more soft and uncertain output. Thus, the transferred model may output [0.1, 0.1, 0.4, 0.3] and [0.1, 0.1, 0.3, 0.4] on two images belonging to the same class in the source model datasets. Although their output probabilities are similar, indicating they are more likely to belong to the same class, their outputs' hard labels are different, and it may cause SAC to fail under hard label setting with transfer learning (A). We have also explained it in the **revised paper**.
>
> **Q2.**
> There is a recent work that proposes a similar solution of measuring pairwise responses to fingerprint models: https://arxiv.org/pdf/2106.12478.pdf. It would be good to cite this paper and point out differences compared to the proposed approach.
>
> **A2.**
> Thanks for the advice and we have cited and compared this paper in the **revised paper**. This paper [C] proposes to generate the image pairs using the matching of the representation layers. Then it calculates the ratio of image pairs to be in the same class in the suspect model as a measurement to judge whether it transfers from the teacher model. It performs well against transferring attacks under different settings. Different from Teacher Model Fingerprinting, our method does not use the paired data (two images with similar features), and we use dozens or hundreds of samples, which belong to different labels, to form the correlation matrix. Our samples do not need to have similar features in the representation layers and thus, we do not need to know which part of the model is frozen and reused by the transferred model. Furthermore, we use the misclassified samples or data augmented samples as the model input and our framework can be applied more generally against different model stealing attacks, such as model extraction, pruning, adversarial training, fine-tuning, and transfer learning.
>
> [C] Chen et al. Teacher Model Fingerprinting Attacks Against Transfer Learning. USENIX 2022.

---

### Official Review · Reviewer_eWaP · 2022-07-11

**Rating:** 5
**Confidence:** 3
**Soundness:** 3 good
**Presentation:** 2 fair
**Contribution:** 3 good

**Summary:**

This paper proposes a method to defend against model stealing attacks. This
method is based on the mean correlation among the selected samples. It provides
two ways to select samples: 1. Finding wrongly classified normal samples. 2.
Selecting mixed samples via CutMix. Experiments on four datasets demonstrate the effectiveness of the proposed method.


**Questions:**

* Could you provide more discussion or analysis about why the proposed method
works?

**Limitations:**

This paper has discussed the limitations and potential negative impacts.


**Strengths And Weaknesses:**

#Strengths

* Interesting topic and good motivation.

* SAC-m has high efficiency.

#Weaknesses

* It lacks solid explanations about why the proposed method works.

* Generalizability to different source models is not clear.

* Comparisons with some related works are missing.

#Detailed comments

* This paper does not provide solid explanations about why the proposed method
works. More discussion or analysis would make the intuition behind the proposed
method more convincing.

* Comparisons with some related works are missing. This paper claims that the
proposed method outperforms previous methods. However, it lacks comparisons with
two related works [1,2]. Therefore, it is hard to say if the proposed method
achieves SOTA performance.

* Table 3 is hard to read. The title of Table 3 mentions "accuracies under
different source models", but it seems there is only one source model.
Therefore, this method's generalizability to different source models is also not
clear.

[1] Li et al. Defending against model stealing via verifying embedded external features. AAAI 2022.

[2] Chen et al. Copy, right? a testing framework for copyright protection of deep learning models. IEEE S&P 2022.

---

> ### Author Response · Authors · 2022-08-02
> **Response to Reviewer eWaP (Part 1)**
>
> Thank you for the detailed reviews, as well as the suggestions for improvement. And we hope to resolve some of your concerns in the following comments.
>
> **Q1.** This paper does not provide solid explanations about why the proposed method works. More discussion or analysis would make the intuition behind the proposed method more convincing.
>
> **A1.**
> Intuitively, samples with similar outputs in the source model are more likely to have similar outputs in the stolen models. Sample correlation (SAC), as a matrix to calculate the correlation of the model's outputs on the specific input samples, can well depict the model's behavior on these samples, and thus can be a unique characteristic of the model. In particular, we can employ the correlation difference between the source model and the suspect model as the indicator to detect the stolen model. The experiment also validates that sample correlation can be well preserved under different model stealing attacks. Although SAC has high effectiveness and good performance, the performance of SAC is still affected by the common knowledge shared by the models on the same task. In other words, the outputs for different models, including the source model, the irrelevant models, and the stolen models on the correctly classified samples are similar, and this will affect the defender to identify the stolen models. Thus, to get rid of the influence of the common knowledge, we propose to use the wrongly classified samples as the model input and calculate the samples' correlation. To avoid the attacker escaping our detection with adversarial training or adversarial extraction, we choose to use wrongly classified normal samples or data augmented samples (CutMix augmented samples) as the model input. Without the common knowledge shared by most models, SAC can be an effective indicator, and SAC-w and SAC-m outperform former model IP protection methods, including IPGuard [19, ICCCS2021], CAE [6, ICLR2021], and EWE [4, USENIX2021].

---

> > ### Author Response · Authors · 2022-08-02
> > **Response to Reviewer eWaP (Part 2)**
> >
> > **Q2.** Comparisons with some related works are missing. This paper claims that the proposed method outperforms previous methods. However, it lacks comparisons with two related works [A,B]. Therefore, it is hard to say if the proposed method achieves SOTA performance.
> >
> > [A] Li et al. Defending against model stealing via verifying embedded external features. AAAI 2022.
> >
> > [B] Chen et al. Copy, right? a testing framework for copyright protection of deep learning models. IEEE S&P 2022.
> >
> > **A2.** Thanks for the advice and we have added the discussions about both papers in the **revised paper**.  VEF [A] proposes to detect the stolen models based on their gradients on the specific style-transfer samples with an MLP classifier. VEF, similar to watermarking, needs to involve in the training process, and it needs the white box access to the suspect model, including the suspect model's architecture and its gradients on the specific style-transfer samples. Although our experiments are based on a black-box setting, VEF can only be conducted in a white box setting, and thus we do VEF experiments in a white box setting. In general, there are two settings for the model IP protection, the white-box setting and the black-box setting. In the black-box setting, the defender can only get access to the suspect models' output, which can be applied more easily and widely. Furthermore, DeepJudge [B] proposes to use the robustness distance (RobD) and the neuron output distance to fingerprint the source model. And, in DeepJudge, there are two settings, the black-box setting, and the white-box setting. We conduct experiments using DeepJudge in a black-box setting. According to the paper, in the black-box setting of RobD, same as IPGuard [19, ICCCS2021], DeepJudge generates the adversarial examples and detects the stolen models using adversarial examples' attack success rate.  Under the same setting, we reuse our experiment results of IPGuard for the black-box setting DeepJudge. Then we show the results of our method and these compared methods as follows:
> >
> > | IP protection |Finetune-A | Finetune-L| Pruning |Extract-L | Extract-P | Extract-Adv| Transfer-A |Transfer-L |
> > | ------- | ------- |------- | -------  |------- | ------- |------- | -------  |-------  |
> > | VEF [A] |1.0 | 1.0| -- |0.86 | 0.68 | 0.86| x |x |
> > | DeepJudge(RobD) [B] |1.0 |1.0| 1.0 |0.81 | 0.80 | 0.52| x |x |
> > | SAC-w |1.0 |1.0| 1.0 |1.0 | 1.0 | 1.0 |1.0 | 1.0 |
> > | SAC-m |1.0 |1.0| 1.0 |0.99 | 1.0 | 0.92 |1.0 | 1.0 |
> >
> > where x represents the model IP protection methods can not detect this kind of model stealing attacks, and -- represents because of the lack of part of the watermark injecting code on Github, we did not do this experiment (Only part of the code is not released, our experiments of VEF are based on the official code from Github).
> >
> > All the experiment results mentioned above are based on CIFAR10 with the same setting as our experiments in Table 1.
> > The only difference is, in VEF [A], we use WideResNet as the source and the stolen models' architectures to directly use the well-trained source and classification models in the source code on Github, where the source model is based on WRN28 and the extracted models are based on WRN16. Both of our methods---SAC-w and SAC-m---only need the output (black-box) access to the victim model and do not need to be involved in the training process, to achieve the SOTA performance.
> >
> > **Q3.**
> > Table 3 is hard to read. The title of Table 3 mentions "accuracies under different source models", but it seems there is only one source model. Therefore, this method's generalizability to different source models is also not clear.
> >
> > **A3.**
> > Thanks for your advice and we have clarified the unclear statement of the name of Table 3 in the **revised paper**. Besides, we have considered different source model situations in our supplementary materials (Table 6 in Appendix). We considered both VGG and ResNet as the source model, and the experiments demonstrated that our sample correlation based fingerprinting method (SAC) performs well among different source model architectures or datasets.

---

> > > ### Comment · Reviewer_eWaP · 2022-08-09
> > > **Thanks for your response**
> > >
> > > Thank you very much for your efforts in addressing these concerns. I maintain my rating and lean to accept this paper.

---

> > > > ### Author Response · Authors · 2022-08-09
> > > > **Thanks again for your support, the detailed reviews, and the suggestions for improvement!**
> > > >
> > > > Thanks again for your support, the detailed reviews, and the suggestions for improvement!

---

> ### Author Response · Authors · 2022-08-09
> **Thanks again for your thoughtful review. Does our response address your questions? We would appreciate the opportunity to engage further if needed.**
>
> Dear reviewer,
>
> Thanks again for your thoughtful review. Does our response address your questions? We would appreciate the opportunity to engage further if needed. We also kindly ask you to consider stronger support for the paper if your concerns have been addressed. Thanks!

---

### Meta-Review · Area_Chair_oAXF · 2022-08-21

**Recommendation:** Accept
**Confidence:** Certain

**Metareview:**

The reviewers agreed that the proposed method and validation overall are a good contribution.

We urge the authors to update their paper to reflect the discussed clarifications, e.g., regarding the threat models in use.



**Award:**

No

---

### Decision · Program_Chairs · 2022-09-14

Accept